# Romantic Attachment, Relationship Satisfaction, Internalized Sexual Stigma, and Motives for Parenthood in Italian Lesbian Women and Gay Men

**DOI:** 10.3390/ijerph20146381

**Published:** 2023-07-17

**Authors:** Massimiliano Sommantico, Marina Lacatena, Ferdinando Ramaglia

**Affiliations:** Dynamic Psychology Laboratory, Department of Humanities, University of Naples Federico II, Via Porta di Massa 1, 80133 Naples, Italy; lacatenamarina@gmail.com (M.L.); ferdinandoramaglia@gmail.com (F.R.)

**Keywords:** lesbian/gay, romantic attachment, internalized sexual stigma, relationship satisfaction, motives for parenthood

## Abstract

This study examines romantic attachment, internalized sexual stigma, relationship satisfaction, and motives for parenthood in a sample of 313 Italian lesbian women (47.9%) and gay men (52.1%) aged 18–71 years (*M* = 36.2; *SD* = 11.9) and in same-sex relationships. The following hypotheses were tested: that romantic attachment is positively correlated with internalized stigma and motives to not have children, while it is negatively correlated with relationship satisfaction; that relationship satisfaction is negatively correlated with internalized stigma and motives for parenthood; that internalized stigma is negatively correlated with motives for parenthood; and that relationship satisfaction mediates the relationships between romantic attachment and motives for parenthood and between internalized stigma and motives for parenthood. The results strongly support the hypotheses. Furthermore, the results indicate that the lesbian participants reported lower levels of avoidance and internalized stigma and higher levels of relationship satisfaction and motives to not have children, and the participants in civil unions reported lower levels of anxiety and internalized stigma and higher levels of relationship satisfaction and motives to not have children. Taken together, our findings contribute to the growing body of research on LG parenthood and may inform social policy and psychological support for LG individuals pursuing parenthood.

## 1. Introduction

Despite the extension of civil union rights to same-sex couples in 2016 in Italy (Law 20, May 2016, n.76) [1], individuals belonging to sexual minoritiescontinue to face additional obstacles compared to heterosexual individuals on their path to parenthood, mainly due to a legal system that prevents them from adoption, assisted reproduction, or surrogacy. However, beyond these legal barriers, lesbian and gay (LG) individuals face sexual minority stressors ranging from discriminatory experiences to internalized expectations of events and conditions of prejudice and discrimination regarding sexual orientation that may shape their parental aspirations [2,3,4].

In 2012, a survey by the Italian National Institute of Statistics (ISTAT) [5] on the prevalence of LG individuals in Italy and on attitudes toward them revealed that 73% of interviewees condemned discriminatory behaviors against LG people and approved of their cohabitation and marriage. Nonetheless, only a small percentage of respondents (20%) were favorable to adoption by same-sex couples, and a smaller percentage felt that it is not acceptable for LG individuals to hold specific jobs such as physician, politician, and elementary school teacher. The survey also highlighted the significant levels of discrimination perceived by LG people. Finally, the number of same-sex couples who defined their relationships “like a family” was 7.513, but it is highly possible that many individuals did not want to declare their status because of the remaining discriminatory and stigmatizing behaviors toward them, as affirmed by LG associations.

These data were confirmed in 2019 by the European Union Agency for Fundamental Rights, which found an increase in the perception of discrimination based on sexual orientation (43% compared to 37% in 2012) in EU countries, leading the European Commission to endorse the LGBTIQ Equality Strategy 2020–2025 in 2020. In particular, Italy is one of the few European Union countries lacking an equality agenda on issues relating to sexual minorities’ rights. Furthermore, according to the European Region of the International Lesbian, Gay, Bisexual, Trans & Intersex Association (ILGA), Italy received a score of 25 on a scale ranging from 0 to 100 with respect to human rights for individuals belonging to a sexual minority [6].

Finally, although there has been some progress in Italy regarding the acceptance of homosexuality, the Catholic religion and the general “don’t ask, don’t tell” attitude toward the LG population [7] have an important influence on negative attitudes toward same-sex parenthood.

### 1.1. Romantic Attachment, Internalized Stigma, and Relationship Satisfaction in LG People

In the international literature examining attachment patterns, romantic attachment has been categorized into two continuous dimensions: avoidance (characterized by the minimization of attachment needs and an individual’s discomfort in close relationships) and anxiety (characterized by the hyperactivation of the attachment system) [8,9], which are strictly associated with relationship quality and satisfaction in both heterosexuals and LG people [10,11]. Evidence of the links between attachment patterns and romantic relationship quality and satisfaction in same-sex couples has been found by several researchers [12,13,14,15,16,17,18], generally indicating a significant positive association between high levels of attachment-related anxiety and avoidance and low levels of relationship satisfaction. This is especially true for two Italian studies conducted in 2020 and 2021 [15,16], whose samples were recruited through the same channels (Italian LG associations, LG listservs, and LG research centers) as the current study.

Indeed, as indicated by Starks and Parsons [12], the “adult attachment style is relevant to the sexual lives of partnered gay men. Secure attachment, in terms of consistent, interdependent, and confident style of relating in a relationship, was associated with higher levels of sexual relationship quality. Securely attached individuals reported the highest levels of sexual communication” (p. 113). Similarly, Wright [14] reported data indicating that “individuals with lower levels of avoidant and anxious attachment tend to be more satisfied in their sexual and romantic relationships compared to those individuals with higher levels of avoidant and anxious attachment” (p. 445).

Furthermore, despite some conflicting results [19], several studies have indicated a strong connection between stress-related variables, such as perceived and internalized sexual stigma, and insecure attachment, but they have also indicated that stress related to discrimination and stigmatization negatively affects the functioning of same-sex couples [20,21,22,23]. Indeed, results from Tognasso and colleagues [15] clearly indicate that “attachment anxiety and avoidance were positively associated with internalized homonegativity” (p. 7).

The internalization of society’s negative ideology and attitudes toward sexual minorities consistently interferes with the relationship satisfaction of same-sex couples, as demonstrated by recent studies and meta-analyses [24,25,26,27] highlighting that the effects of internalized stigma on relationship quality indicators present a greater magnitude of influence than those of perceived stigma. These data confirm that perceived and internalized stigma remain forms of chronic stress that require further sustained attention from researchers interested in understanding the impact of minority stress [28] on the well-being of LG people as well as on their romantic relationships. Indeed, minority stress deprives LG individuals of validation and support for their couple relationships; in turn, this can not only reduce a couple’s ability to cope with stressors but also decrease the commitment, investment, and satisfaction in a romantic relationship [29].

### 1.2. Motives, Desires, and Intentions for Parenthood in LG People

Regarding motives, desires, and intentions for parenthood among individuals belonging to a sexual minority, the international literature has shown mixed results. While in some studies, no significant differences emerged between individuals belonging to a sexual minority and heterosexuals with respect to motives for and against having children and regarding motives for and against becoming a parent [30,31,32,33], other studies highlighted lower levels of desire for parenthood in individuals belonging to a sexual minority compared to heterosexuals [34,35,36,37] by interpreting this data in light of their sexual minority status in the context of societal heterosexism [38,39]. Moreover, legal protections for same-sex couples vary by national context, yet similar patterns of fewer motives and lower levels of desire for parenting among individuals belonging to a sexual minority when compared to heterosexuals have been observed in different international contexts [40,41,42,43,44,45,46]. However, international research has clearly indicated a significant influence of minority stressors (such as the expectation of rejection, experiences of discrimination/prejudice, decisions concerning disclosure/concealment, and internalized stigma/negative views of one’s LG identity or of having a same-sex partner) on the expectations individuals belonging to a sexual minorityhave of achieving parenthood, as well as their motives and desires for parenthood [3,46,47,48,49]. For example, in a recent study, Dorri and Russell [46] found that proximal minority stressors predicted negative parenting expectation gaps, while studies on the Italian LG population [2,3] have clearly indicated that higher levels of internalized homophobia were associated with lower levels of parenting desire in lesbian women, and that felt stigma predicted lower levels of parenting desire and likelihood of parenting in gay men.

Finally, several studies identified a series of challenges encountered by same-sex couples in their process toward parenthood, from social stigma and legal biases in the recognition of their suitability as parents, as well as in their recognition as parents, to difficulties in gaining support and validation for their parental goals from the their families of origin and communities [50,51,52,53,54].

### 1.3. Aim and Hypotheses

In light of the aforementioned literature, the present study aims to analyze the relationships between romantic attachment, relationship satisfaction, internalized sexual stigma, and motives for parenthood in Italian lesbian women and gay men. To this aim, we tested the following hypotheses: that avoidance and anxiety are significantly negatively correlated with relationship satisfaction and significantly positively correlated with internalized sexual stigma and motives to not have children (H1); that relationship satisfaction is significantly negatively correlated with internalized sexual stigma and motives for parenthood (H2); that internalized sexual stigma is significantly negatively correlated with motives for parenthood (H3); and that relationship satisfaction mediates the relationships between romantic attachment and motives for parenthood and between internalized sexual stigma and motives for parenthood (H4).

## 2. Methods

### 2.1. Procedure and Participants

Participants were recruited on Internet forums and through advertisements on social media (LG Associations, LG listservs, and LG research centers), according to the following criteria: participants must identify as lesbian or gay, be over the age of 18 years, and be in a stable relationship (perceived as a healthy and secure relationship, characterized by mutual respect and a strong commitment to each other) lasting at least 6 months. Participation in the study provided no financial incentive and was anonymous and voluntary. Included in the ads disseminated was a link to an online survey (provided through the online platform Qualtrics) that was filled out by 325 people. Information about the purpose, rationale, and procedures of the study was stated on the first page of the survey, where participants also had to provide informed consent. A basic demographic questionnaire was completed on the second page of the survey, while the following pages of the survey consisted of a presentation of four different instruments in the following order: (1) the Experiences in Close Relationships—Revised (ECR-R) scale [55], (2) the Gay and Lesbian Relationship Satisfaction Scale (GLRSS) [56], (3) the Measure of Internalized Sexual Stigma for Lesbians and Gay Men (MISS-LG) [57], and (4) the Motives Towards Parenthood Scale (MTPS) [58] (for a detailed description of the measures, see the “Instruments” section).

The final sample (excluding 12 participants who do not meet the criteria) was composed of 313 participants aged 18–71 years (*M* = 36.2; *SD* = 11.9). Lesbian women made up 47.9% of the sample, and gay men comprised 52.1%. Regarding the participants’ relationship status, 42.2% were cohabiting and 19.8% were in a civil union. The average length of the relationship was 81.4 months (*SD* = 81.9). Finally, 37.4% of the participants had a high school diploma (compared to 64.4% in the Italian population), 40.3% had a bachelor’s degree (compared to 21% in the Italian population), and 16.9% had a postgraduate degree (compared to 12% in the Italian population).

### 2.2. Instruments

#### 2.2.1. Basic Demographic Questionnaire

A basic demographic questionnaire collected information regarding age, gender identity, relationship status (partnered/cohabitation/civil union), relationship length, and educational level.

#### 2.2.2. Experiences in Close Relationships–Revised (ECR-R)

The ECR-R [55,59] is a 36-item self-report questionnaire that provides a dimensional evaluation of current attachment strategies in romantic relationships on two subscales: (1) *Avoidance* (AVO—18 items; e.g., “I prefer not to be too close to romantic partners”) and (2) *Anxiety* (ANX—18 items; e.g., “Sometimes romantic partners change their feelings about me for no apparent reason”). Participants are asked to indicate their feelings regarding their romantic relationship according to a Likert-type scale ranging from 1 (*Totally false*) to 7 (*Totally true*). The ECR-R has demonstrated excellent psychometric properties, with Cronbach’s *α* = 0.90 or higher [55,59], and its internal reliability was excellent in the present study (Cronbach’s *α* was 0.91 for AVO and 0.87 for ANX).

#### 2.2.3. Gay and Lesbian Relationship Satisfaction Scale (GLRSS)

The GLRSS [56,60] is a 24-item self-report questionnaire that measures relationship satisfaction in lesbian and gay respondents on 2 subscales: (1) *Relationship Satisfaction* (RS—16 items; e.g., “Our differences of opinion lead to shouting matches”) and (2) *Social Support* (SS—8 items; e.g., “I feel as though my relationship is generally accepted by my friends”). Participants are asked to express their agreement according to a Likert-type scale ranging from 1 (*Strongly disagree*) to 6 (*Strongly agree*). The GLRSS has demonstrated good psychometric properties, with a Cronbach’s *α* value ranging from 0.72 to 0.83 [56,60], and its internal reliability was good in the present study (Cronbach’s *α* was 0.77 for RS, 0.83 for SS, and 0.81 for the total score).

#### 2.2.4. Measure of Internalized Sexual Stigma for Lesbians and Gay Men (MISS-LG)

The MISS-LG [57] is a 17-item self-report questionnaire measuring internalized sexual stigma on 3 subscales: (1) *Social Discomfort* (SD—7 items; e.g., “I don’t say to my friends that I’m lesbian/gay, because I’m afraid of losing them”); (2) *Sexuality and Affectional Relationships* (SAR—5 items; e.g., “After homosexual intercourse I feel a strong sense of uneasiness”); and (3) *Identity* (I—5 items; e.g., “The thought of being lesbian/gay makes me feel depressed”). Participants are asked to express their agreement according to a Likert-type scale ranging from 1 (*Totally disagree*) to 5 (*Totally agree*). The MISS-LG has demonstrated good psychometric properties, with a Cronbach’s *α* value ranging from 0.72 to 0.83 [57], and its internal reliability was good in the present study (Cronbach’s *α* was from 0.86 for SD, 0.78 for SAR, 0.86 for I, and 0.90 for the total score).

#### 2.2.5. Motives towards Parenthood Scale (MTPS)

The MTPS [58] is a 30-item self-report instrument that produces scores on four first-order subscales. (1) *Emotional Enrichment* (EE—8 items; e.g., “To have someone to take care of”) and (2) *Social Recognition* (SR—8 items; e.g., “To continue the family name”) constitute the second-order subscale *Motives to Have Children* (MHC); (3) *Lifestyle Interference* (LI—9 items; e.g., “I would have less time to spend with my partner”) and (4) *Anticipation of Problems* (AP—5 items; e.g., “The child might not be healthy”) constitute the second-order subscale *Motives to not have Children* (MNHC). Participants are asked to express their agreement according to a Likert-type scale ranging from 1 (*I completely disagree*) to 6 (*I completely agree*). The Italian version of the MTPS was obtained by performing a translation-back-translation, according to the recommendations of the literature on the cross-cultural adaptation of assessment instruments [61]. The MTPS has demonstrated good psychometric properties, with Cronbach’s *α* ranging from 0.74 to 0.87 [58], and its internal reliability was good in the present study (Cronbach’s *α* was 0.85 for EE, 0.71 for SR, 0.88 for MHC, 0.88 for LI, 0.77 for AP, and 0.89 for MNHC).

### 2.3. Data Analyses

The survey data were entered into an SPSS 26.0 [62] database and checked/verified by project staff for accuracy. The instruments’ internal reliabilities were computed via Cronbach’s *α,* which was considered satisfactory at values greater than 0.70 [63]. Correlations were computed via Pearson’s coefficient (small association 0.10 < *r* > 0.29; medium association 0.30 < *r* > 0.49; and large association *r* > 0.50 = large association; *p* < 0.05). Group differences were verified via an ANOVA (*p* < 0.05), and *η*^2^ (small ≥ 0.01; medium ≥ 0.059; large ≥ 0.138) [64] was used to measure effect sizes. Mediation analyses were computed through the PROCESS 4 macro tool for SPSS [65], examining both direct and indirect effects via bootstrapping methods to estimate bias-corrected asymmetric confidence intervals (CIs) with 5000 resamples with replacement (CIs not inclusive of zero indicate significant effects) [66].

## 3. Results

### 3.1. Descriptive Statistics

The means, standard deviations, and Cronbach’s *α* values for the four instruments are presented in Table 1. The means were: Avoidance = 2.1 (*SD* = 1.0), Anxiety = 3.6 (*SD* = 1.1), GLRSS = 100.1 (*SD* = 17.6), MISS-LG = 1.5 (*SD* = 0.5), MHC = 2.5 (*SD* = 0.7), and MNHC = 2.6 (*SD* = 0.8).

### 3.2. Correlation and Group Differences

The results for the total sample (see Table 2) and results stratified for gender (see Table 3) indicated that the age of participants was significantly positively correlated with relationship length and significantly negatively correlated with Avoidance (only in the subsample of lesbian women), Anxiety, GLRSS (in the total sample and in the subsample of lesbian women), and MNHC (in the total sample and in the subsample of lesbian women), while relationship length was significantly positively correlated with GLRSS and significantly negatively correlated with Avoidance (only in the subsample of lesbian women), Anxiety, MISS-LG (only in the subsample of gay men ), and with MNHC (only in the subsamples). Moreover, results for the total sample as well as results stratified for gender indicated that Avoidance was significantly negatively correlated with GLRSS, while it was significantly positively correlated with MISS-LG and MNTC. The results also indicated that Anxiety was significantly negatively correlated with GLRSS and significantly positively correlated with MISS-LG.

These findings strongly support H1, thus suggesting that higher levels of avoidance are significantly correlated with lower levels of relationship satisfaction, as well as with higher levels of internalized sexual stigma and motives to not have children, and that higher levels of anxiety are significantly correlated with lower levels of relationship satisfaction and with higher levels of internalized sexual stigma.

Furthermore, the results for the total sample, as well as results stratified for gender, indicated that GLRSS was significantly negatively correlated with MISS-LG and significantly positively correlated with MNTC, while it was not significantly correlated with MHC. These findings strongly support H2, thus suggesting that higher levels of relationship satisfaction are significantly correlated with lower levels of internalized sexual stigma and with higher levels of motives to not have children.

Finally, the results for the total sample and the results stratified for gender indicated that MISS-LG was significantly positively correlated with MNHC. These findings do not support H3, thus suggesting that higher levels of internalized sexual stigma are significantly correlated with lower levels of motives to not have children.

The ANOVA omnibus test showed statistically significant gender differences. Indeed, lesbian women showed lower levels of Avoidance (*F*_1, 312_ = 5.19, *p* < 0.05; *η*^2^ = 0.02) (*M_L_* = 2.0; *M_G_* = 2.2) than gay men, also reporting higher scores on the GLRSS (*F*_1, 312_ = 4.69, *p* < 0.05; *η*^2^ = 0.01) (*M_L_* = 102.1; *M_G_* = 98.2), lower levels on the MISS-LG (*F*_1, 312_ = 4.63, *p* < 0.05; *η*^2^ = 0.01) (*M_L_* = 1.4; *M_G_* = 1.5), and higher levels of MNHC (*F*_1, 312_ = 3.91, *p* < 0.05; *η*^2^ = 0.01) (*M_L_* = 2.7; *M_G_* = 2.5).

The ANOVA omnibus test and Tukey post-hoc tests also showed statistically significant relationship status differences. Indeed, participants in civil unions reported lower scores than partnered or cohabiting partners on Anxiety (*F*_2, 311_ = 7.26, *p* < 0.01; *η*^2^ = 0.05) (*M_I_* = 3.9, *M_II_* = 3.6, and *M_III_* = 3.2) and the MISS-LG (*F*_2, 311_ = 9.85, *p* < 0.01; *η*^2^ = 0.06) (*M_I_* = 1.6, *M_II_* = 1.5, and *M_III_* = 1.2), while they reported higher scores on the GLRSS (*F*_2, 311_ = 11.14, *p* < 0.01; *η*^2^ = 0.07) (*M_I_* = 97.0, *M_II_* = 98.6, and *M_III_* = 109.1) and higher levels of MNHC (*F*_2, 311_ = 5.96, *p* < 0.01; *η*^2^ = 0.04) (*M_I_* = 2.5, *M_II_* = 2.6, and *M_III_* = 2.9).

No statistically significant differences were found regarding participants’ educational level.

### 3.3. Mediation Analyses

Based on the previous results, we explored the direct and indirect effects of avoidance and internalized sexual stigma on motives to not have children through the variable of relationship satisfaction. We found only indirect effects, reported in Table 4.

Regarding the total sample, the coefficient of the indirect effect for Avoidance was 108.27 (95% CI [−0.20, −0.05]), while the coefficient of the indirect effect for internalized sexual stigma was 122.13 (95% CI [−0.30, −0.19]). Regarding the subsample of lesbian women, the coefficient of the indirect effect for Avoidance was 110.23 (95% CI [−0.22, −0.04]), while for internalized sexual stigma, it was 121.40 (95% CI [−0.32, −0.14]). Finally, regarding the subsample of gay men, the coefficient of the indirect effect for avoidance was 105.69 (95% CI [−0.27, −0.10]), while for internalized sexual stigma, it was 122.55 (95% CI [−0.33, −0.18]). These findings strongly support H4, thus indicating that avoidance, internalized sexual stigma, and relationship satisfaction are associated with motives to not have children. Moreover, the positive coefficients between motives to not have children and relationship satisfaction in both total samples and subsamples (a(313) = 3.89, *p* < 0.01; a(150) = 4.14, *p* < 0.01; a(163) = 3.35, *p* < 0.05), as well as the negative coefficients between motives to not have children and internalized sexual stigma (a(313) = 14.73, *p* < 0.01; a(150) = 13.44, *p* < 0.01; a(163) = 15.67, *p* < 0.01), strongly support Hypothesis 2, thus indicating that participants reporting greater motives to not have children also reported higher levels of relationship satisfaction and of internalized sexual stigma.

## 4. Discussion

The present study aimed to analyze the relationships between romantic attachment, relationship satisfaction, internalized sexual stigma, and motives for parenthood in Italian lesbian women and gay men in romantic relationships.

According to the international literature on well-being and attachment in same-sex couples [12,15,16,22,24,36,67,68,69,70,71], and in our sample, participants with secure attachment patterns showed lower levels of internalized sexual stigma and higher levels of relationship satisfaction, thus indicating the significant role of the attachment system in influencing relationship satisfaction. Furthermore, and consistent with recent research [72,73], in our sample, participants reporting higher levels of avoidant attachment also reported greater motives to not have children. We can interpret this data by hypothesizing that individuals with avoidant attachment have greater reluctance to establish new relationships and cope with the fear of rejection by avoiding new interactions, such as parenting.

Consistent with previous investigations, our results also indicate that lesbian women reported lower levels of avoidant attachment styles [12,13,15,16,18,69,70], higher levels of relationship satisfaction [13,15,21,60,73,74], and lower levels of internalized sexual stigma than gay men [21,60,75,76,77]. These data can be interpreted considering, on the one hand, that as previously suggested [78], lesbian women are more equally involved in or committed to their couple relationships; on the other hand, the prevalence of strong sexism and traditional gender roles in the Italian context may expose gay men to experiences of sexual stigma more frequently than lesbian women [79,80].

Furthermore, according to the international literature [21,57,76,78,81,82,83], our findings also show that younger participants reported higher levels of internalized sexual stigma in the identity dimension. We can interpret this data by hypothesizing that younger lesbian women and gay men are still engaged in the complex process of constructing their own sexual identities. On the contrary, older participants reported lower levels of motives to not have children. This data is in line with both previous Italian studies that found a positive association between age and parenting intentions [41,49] and with the general Italian trend to have children after the age of 30 [84].

Our findings are also consistent with previous studies indicating the positive effect of the civil union on relationship satisfaction [15,16,21,60,85,86,87,88,89]. Indeed, in our study, participants who were joined in a civil union reported lower levels of anxiety and internalized sexual stigma, while they reported higher levels of relationship satisfaction. These data could be interpreted by hypothesizing that the legitimization of their relationship through a civil union may strengthen a couple’s relationship; this could also be due to a broader support network, as well as to greater visibility. In the same vein, we can interpret findings indicating that the participants in the longest-lasting relationships also reported higher levels of perceived social support regarding their romantic relationships, thus confirming the data that emerged from previous studies and reviews [15,16,21,60,78,90]. Moreover, we cannot exclude the presence of selection effects leading to this finding. Indeed, individuals with lower levels of internalized stigma could be more attractive and could be more selective, seeking others with similar levels of internalized stigma, leading to an outcome in which persons who have succeeded in finding a person willing to engage in a civil union would be more likely to report higher levels of relationship satisfaction.

Regarding motives for parenthood, research findings in the Italian LG population are mixed. While some studies have indicated that lesbian women reported higher desires and intentions for parenthood than gay men [3,41,91,92], thus confirming the findings of the international literature [34,93], other Italian studies [49,92] did not find significant differences between lesbian women and gay men. In the present study, lesbian women reported significantly greater motives to not have children compared to gay men. This data could be interpreted in relation to the higher level of relationship satisfaction reported by lesbian woman subsamples that, as previously indicated, was significantly negatively correlated with motives to have children, especially because of possible life interferences. However, we can also hypothesize that, as already observed in other studies [94,95], the COVID-19 pandemic could have strongly undermined or at least complexified participants’ desires for and motives for parenthood. Moreover, even with respect to this result, it is not possible to exclude selection effects. Indeed, especially for lesbian women, it is possible to hypothesize that they might select partners who have similar motivations for/against parenthood and that agreement on parenting motivations might lead to higher relationship satisfaction.

Further, according to previous Italian and international studies [2,3,32,96], participants reporting fewer motives for parenthood also reported higher levels of internalized sexual stigma. These data confirm that the experience and/or expectation of discrimination and stigmatization, as well as negative attitudes toward same-sex parenthood in Italy, negatively affect desires, motives, and intentions for parenthood [97].

### Strengths, Limitations, and Future Research Directions

To our knowledge, this is the first Italian study analyzing the relationships between romantic attachment, relationship satisfaction, internalized sexual stigma, and motives for parenthood in the LG population. Furthermore, rather than simply assuming similarities between lesbian women and gay men, this study also has the strength of distinguishing and comparing the two groups.

As suggested by the international literature, the LG population is conceptually difficult to define and reach, in part due to the frequent difficulty in participating in studies due to the fear that an individual’s disclosure involves risks of stigmatization. In this sense, even with respect to the sample sizes, which were not very large, online surveys have become increasingly popular as a convenient method of collecting data in the LG population [98]. Furthermore, the use of community-based and snowball sampling, which involve biases such as volunteer bias, as well as the likely greater social connectedness of participants recruited online [99], limits the generalization of the study results. Moreover, having assessed all the study variables using self-report measures may have implied the single-method bias.

Future research could deeply investigate different aspects of parenting motivations in the LG population through qualitative data drawn, for example, from in-depth interviews. Finally, longitudinal research designs might be useful in allowing for causal inferences, which were not possible in our study due to its cross-sectional nature.

## 5. Conclusions

Taken together, our results contribute to a better understanding of the roles played by individual, relational, and situational variables that influence motives, desires, and intentions toward parenthood in LG people [100]; they also inform social policies and the provision of psychological support for LG people pursuing parenthood. In this vein, in light of the negative social and political climate relating to same-sex rights in Italy, which is characterized by persistent heteronormativity, the provision of psychological support for lesbian women and gay men people who have motives, desires, and intentions for parenthood and educational programs aimed at informing schools about the challenges and specifics of same-sex parenthood should be central considerations in social policies, as such agencies could play a significant role in improving the well-being of LG people. Finally, the results of the study may significantly contribute to the implementation of specific clinical interventions, such as counseling services, that specifically target same-sex couples who plan to approach the path of parenthood.

## Figures and Tables

**Table 1 ijerph-20-06381-t001:** Descriptive statistics.

	Lesbian Women		Gay Men		Total Sample		
	(*N* = 150)		(*N* = 163)		(*N* = 313)		
	*M*	*SD*	*M*	*SD*	*M*	*SD*	*α*
**Avoidance**	2.0	0.9	2.2	1.0	2.1	1.0	0.91
**Anxiety**	3.5	1.1	3.7	1.0	3.6	1.1	0.87
**Relationship Satisfaction**	68.5	11.4	63.5	12.3	65.9	12.1	0.77
**Social Support**	33.6	11.2	34.8	9.3	34.2	10.3	0.83
**GLRSS Total Score**	102.1	16.6	98.3	18.3	100.1	17.6	0.81
**Social Discomfort**	1.6	0.8	1.7	0.6	1.7	0.7	0.86
**Sexuality and Affections Relationships**	1.2	0.4	1.3	0.4	1.3	0.4	0.78
**Identity**	1.4	0.6	1.5	0.7	1.4	0.6	0.86
**MISS-LG Total Score**	1.4	0.5	1.5	0.5	1.5	0.5	0.90
**Emotional Enrichment**	3.2	1.0	3.3	0.9	3.2	0.9	0.85
**Social Recognition**	1.7	0.6	1.6	0.6	1.7	0.6	0.71
**Motives to Have Children**	2.5	0.7	2.5	0.7	2.5	0.7	0.88
**Lifestyle Interference**	3.3	1.0	3.0	1.0	3.1	1.0	0.88
**Anticipation of Problems**	1.7	0.7	1.6	0.7	1.7	0.7	0.77
**Motives to not have Children**	2.7	0.8	2.5	0.8	2.6	0.8	0.89

**Table 2 ijerph-20-06381-t002:** Correlations between age, relationship length, and the results of the ECR-R, GLRSS, MISS-LG, and MTPS for the total sample (*N* = 313).

	1	2	3	4	5	6	7	8	9	10	11	12	13	14	15	16	17
**1. Age**	-																
**2. Relationship Length**	0.55 **	-															
**3. Avoidance**	0.08	0.09	-														
**4. Anxiety**	−20 **	−0.14 *	0.04	-													
**5. Relationship Satisfaction**	−0.23 **	0.12 *	−0.31 **	−0.24 **	-												
**6. Social Support**	0.08	0.22 **	0.01	−0.18 **	0.22 **	-											
**7. GLRSS Total Score**	−0.11 *	0.15 *	−0.21 **	−0.27 **	0.82 **	0.74 **	-										
**8. Social Discomfort**	−0.01	−0.08	0.11	0.22 **	−0.21 **	−0.51 **	−0.45 **	-									
**9. Sexuality and Affectional Relationships**	0.04	−0.06	0.07	0.16 **	−0.18 **	−0.29 **	−0.29 **	0.57 **	-								
**10. Identity**	−0.14 *	−0.18 **	0.18 **	0.13 *	−0.18 **	−0.32 **	−0.31 **	0.60 **	0.57 **	-							
**11. MISS-LG Total Score**	−0.05	−0.13 *	0.15 *	0.21 **	−0.23 **	−0.47 **	−0.44 **	0.92 **	0.76 **	0.84 **	-						
**12. Emotional Enrichment**	−0.02	−0.05	−0.03	0.07	−0.13 *	0.01	−0.18 **	0.14 **	0.10	0.03	0.12 *	-					
**13. Social Recognition**	−0.07	0.02	−0.12 *	−0.07	0.25 **	0.14 *	0.25 **	−0.00	−0.04	−0.07	−0.04	0.76 **	-				
**14. Motives to Have Children**	−0.04	−0.02	−0.07	0.02	0.02	0.06	−0.1	0.09	0.05	−0.01	0.06	0.96 **	0.91 **	-			
**15. Lifestyle Interference**	−0.11	0.07	0.20 **	−0.15 *	0.52 **	−0.04	0.59 **	0.32 **	0.23 **	0.25 **	0.32 **	−0.12 *	0.28 **	0.04	-		
**16. Anticipation of Problems**	−0.13 *	−0.00	0.09	−0.07	0.43 **	0.33 **	0.49 **	0.24 **	0.15 **	0.17 **	0.24 **	−0.01	0.27 **	0.11	0.69 **	-	
**17. Motives to Not Have Children**	−0.12 *	0.05	0.18 **	−0.13	0.53 **	0.40 **	0.60 **	0.32 **	0.23 **	0.24 **	0.32 **	−0.09	0.30 **	0.07	0.98 **	0.83 **	-

* *p* < 0.05; ** *p* < 0.01.

**Table 3 ijerph-20-06381-t003:** Correlations between age, relationship length, and the results of the ECR-R, GLRSS, MISS-LG, and MTPS for lesbian women and gay men (lesbian women *N =* 150; gay men *N* = 163).

	1	2	3	4	5	6	7	8	9	10	11	12	13	14	15	16	17
**1. Age**	-	0.56 **	−0.06	−0.19 *	−0.06	0.26 **	0.09	−0.04	0.02	−0.21 **	−0.10	−0.14	−0.12	−0.14	0.05	−0.01	0.03
**2. Relationship Length**	0.56 **	-	−0.02	−0.15 *	0.11	0.33 **	0.24 **	−0.14	−0.08	−0.25 **	−0.20 *	−0.17 *	0.04	−0.09	−0.29 *	0.15	−0.27 **
**3. Avoidance**	−0.23 **	−0.20 **	-	−0.03	−0.21 **	−0.08	−0.18 *	0.07.	0.05	0.19 **	0.15 *	−0.07	−0.13	−0.10	0.18*	0.10	0.17 *
**4. Anxiety**	−22 **	−0.17 *	0.10	-	−0.30 **	−0.21 **	−0.31 **	0.23 **	0.13	0.02	0.16 *	0.16 *	0.01	0.11	−0.14	−0.06	−0.13
**5. Relationship Satisfaction**	−0.40 **	0.40 **	−0.38 **	−0.13	-	0.42 **	0.89 **	−0.24 **	−0.17 **	−0.16 **	−0.24 **	−0.08	0.38 **	0.12	0.58 **	0.48 **	0.59 **
**6. Social Support**	−0.08	−0.08	0.05	−0.18 **	0.07	-	0.79 **	−0.51 **	−0.29 **	−0.43 **	−0.52 **	−0.18 *	0.20 **	−0.03	0.45 **	0.32 **	0.44 **
**7. GLRSS Total Score**	−0.34 *	0.22 **	−0.23 **	−0.21 **	0.74 **	0.72 **	-	−0.42 **	−0.27 **	−0.33 **	−0.42 **	−0.14	0.36 **	0.06	0.62 **	0.49 **	0.63 **
**8. Social Discomfort**	0.01	−0.06	0.13	0.21 **	−0.17 **	−0.52 **	−0.47 **	-	0.58 **	0.55 **	0.90 **	0.18 *	−0.05	0.09	0.28 **	0.19 **	0.27 **
**9. Sexuality and Affectional Relationships**	0.03	−0.09	0.06	0.17 **	−0.13	−0.32 **	−0.30 **	0.56 **	-	0.51 **	0.76 **	0.06	−0.13	−0.01	0.24 **	−0.13	0.23 **
**10. Identity**	−0.07	−0.11	0.15*	0.28 *	−0.18*	−0.23 **	−0.28 **	0.66 **	0.64 **	-	0.83 **	−0.00	−0.18 **	−0.08	0.26 **	0.22 **	0.26 **
**11. MISS-LG Total Score**	−0.01	−0.09	0.15 *	0.25 **	−0.19 *	−0.46 **	−0.44 **	0.94 **	0.75 **	0.85 **	-	0.11	−0.13	0.01	0.31**	0.22 **	0.31 **
**12. Emotional Enrichment**	0.09	0.09	−0.01	−0.02	−0.18 *	−0.13	−0.21 **	0.11	0.13	0.07	0.12	-	0.71 **	0.95 **	−0.09	0.04	−0.06
**13. Social Recognition**	−0.01	−0.03	−0.10	−0.14	0.08	0.08	0.11	0.05	0.07	0.06	0.07	0.81 **	-	0.89 **	0.37 **	0.39 **	0.40 **
**14. Motives to Have Children**	0.06	0.07	−0.04	−0.07	−0.08	−0.05	−0.9	0.09	0.11	0.07	0.10	0.97 **	0.92 **	-	0.09	0.20 *	0.14
**15. Lifestyle Interference**	−0.26 **	−0.20 *	0.20 **	−0.13	0.42 **	0.37 **	0.54 **	0.35 **	0.19 *	0.22 **	0.32 **	−0.13	0.18 *	−0.01	-	0.68 **	0.97 **
**16. Anticipation of Problems**	−0.26 **	−0.22 **	0.07	−0.07	0.35 **	0.35 **	0.48 **	0.29 **	0.16	0.10	0.24 **	−0.05	0.12	0.01	0.69 **	-	0.83 **
**17. Motives to Not Have Children**	−0.28 **	−0.22 **	0.17 **	−0.12	0.43 **	0.39 **	0.56 **	0.35 **	0.19 *	0.20 *	0.32 **	−0.12	0.18 *	−0.01	0.98 **	0.83 **	-

Note: Correlations for lesbian women are below the diagonal; correlations for gay men are above the diagonal. ** p* < 0.05; ** *p* < 0.01.

**Table 4 ijerph-20-06381-t004:** Mediated outcomes on the MNHC, showing indirect effects of avoidance, anxiety, and the MISS-LG through the GLRSS.

	Total Sample (*N* = 313)						Lesbian Women (*N* = 150)						Gay Men (*N* = 163)					
	Avoidance			MISS-LG			Avoidance			MISS-LG			Avoidance			MISS-LG		
Pathway	*b*	*SE*	*p*	*b*	*SE*	*p*	*b*	*SE*	*p*	*b*	*SE*	*p*	*b*	*SE*	*p*	*b*	*SE*	*p*
**a**	−3.89	1.01	0.0001	14.73	1.73	<0.0001	−4.14	1.41	0.004	13.44	2.26	<0.0001	−3.35	1.44	0.021	15.67	2.65	<0.0001
**b**	−0.03	0.00	<0.0001	0.03	0.00	<0.0001	0.03	0.00	<0.0001	0.02	0.00	<0.0001	0.03	000	<0.0001	0.03	0.00	<0.0001
**c** **′**	−0.05	0.04	0.211	−0.11	0.08	0.157	−0.04	0.06	0.514	−0.13	0.11	0.245	−0.05	0.05	0.334	−0.08	0.11	0.460
**i.e.,**	108.27	2.33	<0.0001	122.13	2.74	<0.0001	110.23	3.10	<0.0001	121.40	3.47	<0.0001	105.69	3.50	<0.0001	122.55	4.31	<0.0001
**Total *R*^2^**	0.05		0.0001	0.19		<0.0001	0.05		0.004	0.20		<0.0001	0.03		0.021	0.18		<0.0001

Note: i.e., = indirect effect.

## Data Availability

The data presented in this study are available upon request from the corresponding author.

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
