# Peer review of "Romantic Attachment, Relationship Satisfaction, Internalized Sexual Stigma, and Motives for Parenthood in Italian Lesbian Women and Gay Men"

_ijerph, 2023, doi:10.3390/ijerph20146381_

Round 1

Reviewer 1 Report (Previous Reviewer 1)

 Accept in present form

This manuscript is a resubmission of an earlier submission. The following is a list of the peer review reports and author responses from that submission.

Round 1

Reviewer 1 Report

Thanks to the authors for this work, which I highly appreciate.

Below some minor revision and comment:

- the introduction point  is a bit too long and I would suggest reducing it by summarising in main points (i.e. the part on ISTAT from line 43 to line 56 is really very long) ;

- please delete the "s" of "aims" in line 130;

- please explain what you mean by "stable relationship";

- did you translate the MTPS? If so, did you by chance also do a convergent analysis or was the translation carried out for the present study? (line 211-216);

- it would be interesting if you could also indicate the clinical impact of this research;

Please add the following reference:

Santoniccolo, F., Trombetta, T. & Rollè, L. The Help-Seeking Process in Same-Sex Intimate Partner Violence: a Systematic Review. Sex Res Soc Policy 20, 391–411 (2023). https://doi.org/10.1007/s13178-021-00629-z;

  1.  

Author Response

  1. We have minimized the part on ISTAT, but this part seems to us essential to account for the context in which the research took place and because that context certainly has a specific influence on the variables examined in the study;
  2. We deleted the "s" of "aims" in line 130;
  3. Asking participants to indicate whether they were in a stable relationship, we defined it as a relationship that is perceived as healthy and secure and characterized by mutual respect and a strong commitment to each other, regardless of whether it is a monogamous relationship or open to polyamory;
  4. The translation of the MTPS was carried out for the present study;
  5. We added in the Conclusions some lines regarding the clinical impact of the study;
  6. We added the suggested reference.

Reviewer 2 Report

Method Section

1. How did you operationalize a "stable relationship"? Are these only monogamous relationships or open to polyamorous couples as well?

2. Given that you are presenting data on motives towards parenthood, having information about the household income of the sample will help to further put the results in context. 

3. Add specific numbers of psychometrics from other studies that you reference in your methods section (as opposed to stating that a measure has "demonstrated good or excellent psychometric properties", it's helpful for the reader if you include specifics as to what exactly those psychometric properties were.

4. Specify which PROCESS model was used for your analysis

Author Response

  1. Asking participants to indicate whether they were in a stable relationship, we defined it as a relationship that is perceived as healthy and secure and characterized by mutual respect and a strong commitment to each other, regardless of whether it is a monogamous relationship or open to polyamory;
  2. We have no data regarding household income;
  3. We added data regarding Cronbach’s alpha reported by referenced studies;
  4. We specified that we used Process 4.

Reviewer 3 Report

Thank you for submitting such a thought provoking manuscript. The manuscript read well and was easy to navigate. I am in awe with how much data you were able to collect. Below are a few comments I have. 

2. Line 45- there is an extra parenthesis. 

pg 2 Line 55- 7.513 what?

Line 67 - that sentence should belong to the sentence above since it is not a complete paragraph. 

Line 81- I would define what secure attachment means. 

Line 116-  I would define what minority stressors are. 

The authors should define their variables to give the read the understanding of what is being measured. 

Methods Section. 

I feel that your methods section was great at explaining the surveys and the methods behind recruitment for the study. 

Results.

I would identify which variables were significant and then discuss the type of correlation. There is a lot of data.

Author Response

  1. There is no extra parenthesis in line 45 (see the first parenthesis in line 44);
  2. In line 55 the number 7.513 refers to the same-sex couples who defined their relationship “like a family” (see line 54)
  3. We adjusted the sentence in line 67;
  4. We defined secure attachment in Line 81;
  5. We defined minority stressors in line 116;
  6. We removed the correlation data from the results (since they were already in Tables 2 and 3) and commented on the results with respect to the study hypotheses.